# Improved Quantitative Real-Time PCR Protocol for Detection and Quantification of Methanogenic Archaea in Stool Samples

**DOI:** 10.3390/microorganisms11030660

**Published:** 2023-03-05

**Authors:** Agata Anna Cisek, Iwona Bąk, Bożena Cukrowska

**Affiliations:** 1Department of Pathology, The Children’s Memorial Health Institute, Av. Dzieci Polskich 20, 04-730 Warsaw, Poland; 2Department of Preclinical Sciences, Institute of Veterinary Medicine, Warsaw University of Life Sciences, St. Ciszewskiego 8, 02-786 Warsaw, Poland

**Keywords:** archaea, *mcrA*, methanogens, *Methanobrevibacter*, qPCR, real-time PCR

## Abstract

Methanogenic archaea are an important component of the human and animal intestinal microbiota, and yet their presence is rarely reported in publications describing the subject. One of the methods of quantifying the prevalence of methanogens is quantitative real-time PCR (qPCR) of the methanogen-specific *mcrA* gene, and one of the possible reasons for detection failure is usually a methodology bias. Here, we refined the existing protocol by changing one of the primers and improving the conditions of the qPCR reaction. As a result, at the expense of a slightly lower yet acceptable PCR efficiency, the new assay was characterized by increased specificity and sensitivity and a wider linear detection range of 7 orders of magnitude. The lowest copy number of *mcrA* quantified at a frequency of 100% was 21 copies per reaction. The other validation parameters tested, such as reproducibility and linearity, also gave satisfactory results. Overall, we were able to minimize the negative impacts of primer dimerization and other cross-reactions on qPCR and increase the number of not only detectable but also quantifiable stool samples—or in this case, chicken droppings.

## 1. Introduction

Methanogenic archaea are one of the many natural residents of animal and human intestines [1,2]. They partake in hydrogen sink, a process of the utilization of hydrogen, a byproduct of intestinal microbial fermentation. It is known that, without hydrogen sink, the fermentation itself would slow down, thus depriving the host of various useful nutritious compounds [3]. Moreover, methanogens interact closely with other intestinal microbionts, and their absence may be an indicator of intestinal dysbiosis [1]. Methanogens may also influence their hosts in other ways. On the one hand, some archaea were reported to have a probiotic potential, such as *Methanomassiliicoccus luminyensis* B10 in treating trimethylaminuria (TMAU), i.e., the fish-odor syndrome [4], and *Methanobrevibacter smithii* in treating severe acute malnutrition [5]; on the other hand, species such as *Methanosphaera stadtmanae* may promote inflammation [6].

Of all methanogenic archaea, species of the genus *Methanobrevibacter* are known to be predominant methanogen taxa in the gastrointestinal tracts of animals and humans. In non-rumen animals, they are followed by *Methanosphaera*, *Methanosarcina*, *Methanomassiliicoccus*, and *Methanimicrococcus*. In chickens, *Methanobrevibacter woesei* is the only dominating genus. The rumen microbiota is significantly more diverse in terms of archaeal prevalence, with four main orders, i.e., Methanomicrobiales, Methanosarcinales, Methanobacteriales, and Methanomassiliicoccales [7].

It is estimated that methanogenic archaea account for 0.05–0.8% of the intestinal microbiota in humans, 4% of the rumen microbiota, and 1–2% of the chicken cecal microbiota [7,8,9]. However, since the detection of archaea is challenging, those data may not be precise. For instance, to date, many studies do not report the presence of archaea in the chicken ceca at all [10,11], whilst others say otherwise [12,13]. For this reason, chicken dropping samples were chosen as a model in this study.

It is not known whether the lack of reported archaea in the chicken intestinal microbiome is caused by their actual absence or the methodology limitations of those studies. Such limitations may include insufficient cell lysis during DNA isolation from archaea or an inadequate PCR protocol. The first issue was addressed in our previous study [14]; therefore, here, we decided to take a closer look at the subject of the ‘detection’ of methanogens via real-time PCR.

The detection of methanogenic archaea using real-time PCR is usually performed with the use of primers targeting either the 16S rDNA or the genes involved in methanogenesis, uniquely specific to methanogenic archaea, usually the *mcrA* gene encoding the alpha subunit of methyl-coenzyme M reductase [15]. The first target has a few major drawbacks. The 16S rDNA may not be selective enough to allow for the identification of only the methanogenic archaea and, therefore, for the undertaking of PCR on templates isolated directly from feces. The other drawback is related to the high scattering of the 16S rDNA copy number between the different taxa of archaea [16], which, as a result, does not allow for the quantification of these microbes in samples. Putting these facts together, the 16S rDNA seems the least preferred target for the quantification of methanogens in dropping or stool samples.

The *mcrA* gene, however, is a single-copy gene, which makes it ideal for quantification purposes [15]. The main difficulty in application comes from the fact that methanogenic archaea are very genetically diverse, and, therefore, finding a good location for primers along the *mcrA* gene and making sure that they comply with the rules of proper primer design can be challenging. Here, we addressed this issue and undertook exhaustive efforts in refining the existing protocol of the quantification of methanogenic archaea.

## 2. Materials and Methods

### 2.1. Primer Design

A total of 47 *mcrA* sequences of methanogenic archaea were selected from the NCBI GenBank database. The sequences were aligned by using the webPRANK tool [17] in order to identify the conserved regions (Appendix A). Several potential forward primers were created, and they were checked together with the reverse primer *mcrA*-rev [18] (Table 1) in the Oligo Analyzer [19] and Primer-BLAST [20] tools with regard to their specificity and tendency to form secondary structures.

### 2.2. DNA Templates

The total genomic DNA of 3 genetically and taxonomically diverse reference strains of methanogens, i.e., *Methanobrevibacter woesei* DSM 11979, *Methanococcus maripaludis* DSM 2067, and *Methanomicrobium mobile* DSM 1539, was chosen as positive controls and standards. Moreover, since the BLAST analysis of the *mcrA* sequence demonstrated that *Methanobrevibacter* sp. D5 was the closest to the consensus of 26 methanogens (*Methanobrevibacter* spp. and environmental species; Appendix A) in the binding site of the F3 primer, its 472 bp long *mcrA* gene fragment was picked out as a group representative and the fourth positive control. It was used in the form of a plasmid construct (*mcrA*_MB) carrying the 472 bp long mlas/*mcrA*-rev primer amplicon. For quantification purposes, the *mcrA*-positive (*mcrA*+) plasmid was linearized with the use of a single cutting VspI restriction enzyme (Thermo Fisher Scientific, Waltham, MA, USA). Then, in order to select only the linearized form, it was subjected to agarose gel electrophoresis. A band representing the linearized form was cut out, purified with a Basic DNA Purification Kit (EURx, Gdańsk, Poland), and diluted in a Tris buffer (10 mM Tris HCl, pH 8.5), as were all the other DNAs used in this study. In order to create 4 standard curves, all 3 archaeal DNAs and one *mcrA*+ plasmid were serially diluted 10-fold, from approx. 10^6^ to 10^−1^ genome copies/µL.

The genomic DNA of 21 bacterial strains, i.e., *Enterococcus avium* ATCC 14025, *Enterococcus casseliflavus* ATCC 700327, *Enterococcus raffinosus* ATCC 49464 (courtesy of Dr. habil. Beata Dolka, Institute of Veterinary Medicine, Warsaw University of Life Sciences—SGGW), *Lactobacillus sakei* ATCC 15521 (courtesy of Dr. Ilona Stefańska, Institute of Veterinary Medicine, Warsaw University of Life Sciences—SGGW), *Escherichia coli* ATCC 8739, *Salmonella enterica* subsp*. enterica* serovar Typhimurium ATCC 14028, *Clostridium septicum* ATCC 12464, *Blautia obeum* DSM 25238, *Ruminococcus gauvreauii* DSM 19829, *Helicobacter cinaedi* DSM 5359, *Desulfovibrio piger* DSM 749, *Proteus* sp., *Streptococcus* sp., *Streptococcus* sp. (beta-hemolytic), *Corynebacterium* sp., *Pseudomonas aeruginosa*, *Pasteurella* sp., *Klebsiella* sp., *Staphylococcus* sp. (coagulase-negative), *Porphyromonas* sp., and Bacteroidetes bacterium, was the non-target control of PCR. The strains were purchased or received either in the form of DNA or a bacterial pellet, from which the DNA was isolated with the use of a Genomic Bacteria+ kit (A&A Biotechnology, Warsaw, Poland) according to the producer’s instructions. The concentrations of the controls were measured with a Quantus fluorometer and the QuantiFluor dsDNA System (Promega Corporation, Madison, WI, USA), and they were converted into the number of genome copies per µL by using the Science Primer web tool [22].

Lastly, 20 dropping samples collected from free-range chickens were subjected to DNA isolation with the use of a Genomic Mini AX Bacteria+ kit (A&A Biotechnology, Gdynia, Poland) and to mechanical lysis via sonication in accordance with the protocol described previously [14].

### 2.3. Initial Comparison of the Mlas and mcrA_F3 forward Primers

For unification purposes, the *mcrA*_F3/*mcrA*-rev and mlas/*mcrA*-rev primer pairs were compared with the use of the same reagents routinely used in our lab. The reaction mixture included 10 µL of RT HS-PCR Mix SYBR A (A&A Biotechnology, Gdynia, Poland), 0.5 µM of each primer (Table 1), 1 µL of *mcrA*+ plasmid in 10-fold dilutions, and water with BSA (3.5 µg per reaction) to reach a final volume of 20 µL. The reaction conditions closely resembled the conditions from the original study [18], and they were as follows: initial denaturation at 95 °C for 3 min; 45 cycles comprising denaturation at 95 °C for 30 s; annealing at 55 °C for 45 s; extension at 72 °C for 30 s; and fluorescence acquisition either at 81 °C or 83 °C after 20 s for mcrA_F3/*mcrA*-rev and mlas/*mcrA*-rev primer pairs, respectively. In contrast to the original publication, no additional preincubation at 37 °C or postincubation at 72 °C was implemented.

### 2.4. Gradient PCR

The reaction mixture included 15 µL of RT HS-PCR Mix SYBR A (A&A Biotechnology, Gdynia, Poland), 0.5 µM *mcrA*_F3 and *mcrA*-rev primers (Table 1), 6.24 × 10^4^ of *mcrA*+ plasmid, and water with BSA (3.5 µg per reaction) to reach a final volume of 30 µL. A gradient PCR with an annealing temperature between 54 and 62 °C was performed. The results were visualized using agarose gel electrophoresis.

### 2.5. Selected Validation Parameters of the Assays

The sensitivity and specificity of the assays were evaluated for standards from approx. 10^−1^ to 10^6^ copies per reaction and for at least 7 runs. The intra- and inter-assay variability was tested in triplicate. Melting analyses were performed after every run to check the specificity of the assays.

In the SYBR Green detection approach, the reaction mixture included 10 µL of RT HS-PCR Mix SYBR A (A&A Biotechnology, Gdynia, Poland), 0.5 µM primers (Table 1), 1 µL of DNA, and water with BSA (3.5 µg per reaction) to reach a final volume of 20 µL. A real-time PCR was performed in which the slopes after denaturation and after annealing were set up to the universal value of 2.2 °C/s. The reaction conditions for the mlas/*mcrA*-rev primer pair were as follows: initial denaturation at 95 °C for 3 min, 45 cycles comprising denaturation at 95 °C for 30 s, annealing at 55 °C for 45 s, extension at 72 °C for 30 s, and fluorescence acquisition at 83 °C after 20 s. The reaction conditions for the *mcrA*_F3/*mcrA*-rev primer pair were as follows: initial denaturation at 95 °C for 3 min, 45 cycles comprising denaturation at 94 °C for 20 s, annealing at 60 °C for 20 s, extension at 72 °C for 20 s, and fluorescence acquisition at 81 °C after 20 s.

### 2.6. Detection of Methanogens in Chicken Dropping Samples

The DNA samples isolated from the chicken droppings were tested with the use of the *mcrA*_F3/*mcrA*-rev and mlas/*mcrA*-rev primer pairs as described above, with 100 to 200 ng of DNA used for the tests. The products were separated by using agarose gel electrophoresis in order to double check the specificity of the assays.

## 3. Results

### 3.1. Primer Design

A new forward primer (*mcrA*_F3) was designed approx. 176 bases downstream of the original forward primer (mlas), making future amplicons shorter and, therefore, presumably better for quantification purposes (Table 2).

### 3.2. Initial Comparison of the Mlas and mcrA_F3 Forward Primers

Both primer pairs were compared under very similar temperature conditions and timings as presented in the original study in order to exclude any additional effects on the assay performance. Since the amplicons generated with the use of the *mcrA*_F3/*mcrA*-rev primer pair were shorter and had a slightly lower melting temperature (Tm), the temperature of fluorescence acquisition had to be adjusted—it was set to 81 °C instead of 83 °C.

At an annealing temperature of 55 °C, both primers performed similarly well in terms of Cq values for the five most concentrated dilutions of the *mcrA*+ plasmid (Figure 1). However, there was a profound difference in the shape and height of the amplification curves as the dilution ratio increased. In contrast to the original primer pair, the *mcrA*_F3/*mcrA*-rev pair generated curves that were well-pronounced until the last standard dilution, i.e., 6.24 copies of the *mcrA*+ plasmid per reaction. Since there was a minimal signal from the non-template control (NTC), an optimization of the real-time PCR conditions for the new primer pair was required to keep primer dimerization as minimal as possible.

### 3.3. Gradient PCR

The primer pair *mcrA*_F3/*mcrA*-rev performed best at an annealing temperature (Ta) in the range of 59–60 °C. Amplification at 61 °C and above resulted in increasing primer dimerization (Figure 2).

### 3.4. Selected Validation Parameters of the Assays

At approx. 10^1^ copies per reaction, 100% of the replicates of all four standard DNAs tested positive using the *mcrA*_F3/*mcrA*-rev primer pair, whilst only three standard DNAs tested positive using the mlas/*mcrA*-rev primer pair (Table 3). The only exception—*Methanomicrobium mobile—*had two limits of detection determined for each primer pair separately, i.e., 5.71 × 10^2^ (or 2.76 log10) and 5.71 × 10^1^ (or 1.76 log10) copies per reaction using the original and the *mcrA*_F3/*mcrA*-rev primer pair, respectively.

Moreover, the *mcrA*_F3/*mcrA*-rev primer pair demonstrated a higher amplification specificity than the mlas/*mcrA*-rev primer pair, as it favored the generation of specific products over primer dimers, even at low concentrations of the target DNAs. This was especially noticeable in the case of *Methanomicrobium mobile* and the *mcrA*+ plasmid (Table 3; Figure 3).

Table 4 presents the reproducibility and efficiency of the assays. Only the Cq values for the specific reactions are reported. Efficiency within the acceptable range of 90 to 110% was observed in all cases. Amplification with the *mcrA*_F3/*mcrA*-rev primer pair occurred with lower efficiency, and it was more variable at lower template concentrations.

The dynamic range, referred to as the range of template input generating a linear curve (R^2^ ≥ 0.980) with acceptable efficiency, was larger in the case of the new primer pair. Here, the dynamic range of the real-time PCR assay measured for three standard DNAs (*mcrA*+ plasmid, *Methanococcus maripaludis*, and *Methanomicrobium mobile*) spanned 6 to 7 orders of magnitude, and for *Methanobrevibacter woesei*, it spanned 5 orders of magnitude. In contrast, amplification with the use of the mlas/*mcrA*-rev primer pair generated only two standard curves covering a linear range of 6 orders of magnitude (of the *mcrA*+ plasmid and *Methanococcus maripaludis*). The dynamic range for the remaining two standard DNAs (*Methanobrevibacter woesei* DSM 11979 and *Methanomicrobium mobile*) covered only 5 orders of magnitude.

The R^2^ value was between 0.9934 and 0.9966 for the mlas/*mcrA*-rev primer pair and between 0.9913 and 0.9992 for the *mcrA*_F3/*mcrA*-rev primer pair.

The reproducibility of the assays as defined by the values of standard deviation (SD) and the coefficient of variation (CV%) in the within-run and between-run tests are reported independently for all four tested *mcrA*+ genomes in Table 4.

The Cq values of the generated non-target controls were higher in the case of the new *mcrA*_F3/*mcrA*-rev primer pair, usually well above the value of 30 (Table 5). Moreover, by comparing the Cq values of the bacterial DNA and the Cqs of the last *mcrA*+ dilution points, the comparison favors the new primer pair, as the calculated difference between the Cqs is simply bigger. In contrast, as many as 18 non-target controls had a Cq lower than 30.8, which was the lowest reported value for the last standard dilution point for the mlas/*mcrA*-rev primer pair (Table 4). Moreover, the amplification curve generated, e.g., from *Escherichia coli* by using the original primer pair indicated a strong false-positive result (viewed as a high rate of increase), which was not observed in the case of the *mcrA*_F3/*mcrA*-rev primer pair (Figure 4).

The melting profile of the mlas/*mcrA*-rev amplicons generated from the bacterial DNAs indicated a substantial amplification of the non-specific products from *E. coli* and *Enterococcus avium*. Those amplicons had a Tm within or above the range of specific products, making them measurable in terms of the Cq. In contrast, the tendency to amplify products with a Tm around the specific peak was marginal when the mcrA_F3/*mcrA*-rev primer pair was used (Figure 4).

### 3.5. Methanogen Load in Chicken Dropping Samples

Given that the temperature in the range between approx. 83 and 87 °C was determined experimentally as being *mcrA*-specific, only seven dropping samples could be considered *mcrA*+ using the original mlas/*mcrA*-rev primer pair (Table 6). In contrast, as many as 13 samples fell under the same criteria when the *mcrA*_F3/*mcrA*-rev was used, and another 7 samples were just under the lower limit.

The primer pair mlas/*mcrA*-rev did not amplify any specific product in the remaining 13 dropping samples despite generating low Cq values in all cases but one (sample 8).

## 4. Discussion

Publications bringing up the subject of primer and qPCR validation are sparse, with a strong emphasis being placed on pathogens [23,24,25]. This paper is—to the best of our knowledge—the first one that focuses on methanogenic archaea in dropping samples of chickens.

The primers first designed by Steinberg and Regan [21] were used with success by several authors, providing a great deal of knowledge on the subject of diversity and the prevalence of methanogenic archaea in various environments, i.e., in landfills, wastewater, the gastrointestinal tracts of insects, and human stool samples [26,27,28,29].

However, our very preliminary research showed that, by using these primers together with the different temperature settings reported previously for qPCR by Steinberg and Regan [18] and Lecours et al. [6], methanogens remained undetected in the majority of the tested dropping samples. In our opinion, the reason behind this was the length of the generated *mcrA* amplicons of approx. 470–490 bp. Therefore, we tried to explore the possibility of designing a new assay.

Once the new forward primer was designed, the next step was to compare its performance to that of the original one. In order to exclude the potential influence of any other conditions (such as the annealing temperature) on the primer performance, this test was performed as closely as possible to the conditions described in the original study by Steinberg and Regan [18]. By changing only one primer, we were able to obtain a significant improvement. Since there was some evidence of primer dimerization resulting in the generation of a signal in the NTC, the real-time PCR protocol had to be optimized. As a consequence, the *mcrA*_F3 forward primer—together with the original reverse primer and improved PCR conditions—was able to generate amplicons of approx. 270 bp under more stringent conditions, which, as was later proven, was enough to detect methanogens in almost all tested samples that previously underwent mechanical lysis according to the protocol we designed previously [14]. In the aforementioned study, we used a hydrolysis probe in order to reduce any unspecific signals that could have arisen from the primer dimers [14]; however, the use of this probe might have resulted in the loss of some reads from the methanogenic archaea due to potential mismatches between the probe and its target. Here, we tried a different approach—a four-step real-time PCR with fluorescence acquisition at temperatures just under the point of melting of the specific products. Such changes induced the generation of lower Cq values of the assay.

A higher annealing temperature favored the generation of specific products over primer dimers, even at low concentrations of the target DNAs. Moreover, our assay improved specificity with respect to the cross-reactions with the non-target controls. In the original assay, even the temperature of fluorescence acquisition set to 83 °C was not high enough to prevent one from measuring the Cq of those non-specific products, since their Tm was, in some cases, as high as or even higher than the Tm of the specific ones. Of course, the non-specific reactions using the mlas/*mcrA*-rev primer pair may have occurred due to the fact that the annealing temperature was relatively low, but by analyzing the results from the initial experiment, no further optimization would have given results similar to the primer pair proposed in our study.

Although the Cq values of the standards generated using the *mcrA*_F3/*mcrA*-rev primer pair were generally higher than those generated using the previously published primer pair, the ability to differentiate between the positive and the non-target controls by comparing the Cq values was much higher using the *mcrA*_F3/*mcrA*-rev primer pair. In general, the *mcrA*_F3/*mcrA*-rev assay led to the amplification of three out of four positive controls and all dropping samples with a higher sensitivity. Both the intra- and inter-assay CVs for the standard DNAs were satisfactory low. This assay was also more specific than the mlas/*mcrA*-rev assay, and it had a reproducible limit of detection of approx. 21 to 57 copies of target DNA per reaction depending on the DNA template used.

Steinberg and Regan [18] reported that they were able to achieve a limit of detection of approx. 415 copies per reaction with the use of the mlas/*mcrA*-rev primer pair. Our results show that the limit of detection generated by the same primers is in the range of 21 to 571 copies per reaction, which is in line with the original study.

## 5. Conclusions

The current results present a novel qPCR approach in the detection of methanogenic archaea using the *mcrA*_F3/*mcrA*-rev primer pair. Although the proposed protocol was not able to entirely eliminate non-specific reactions (cross-amplification), most validation parameters improved significantly. Moreover, the shortening of the amplicon allowed for a more accurate quantification of methanogenic archaea in dropping and stool samples, especially those that underwent mechanical lysis during DNA isolation. In the long run, the proposed protocol may improve the detection rates of methanogens in human and animal gastrointestinal tracts and contribute to a better understanding of the role of archaea in the health and disease of their hosts.

## Figures and Tables

**Figure 1 microorganisms-11-00660-f001:**
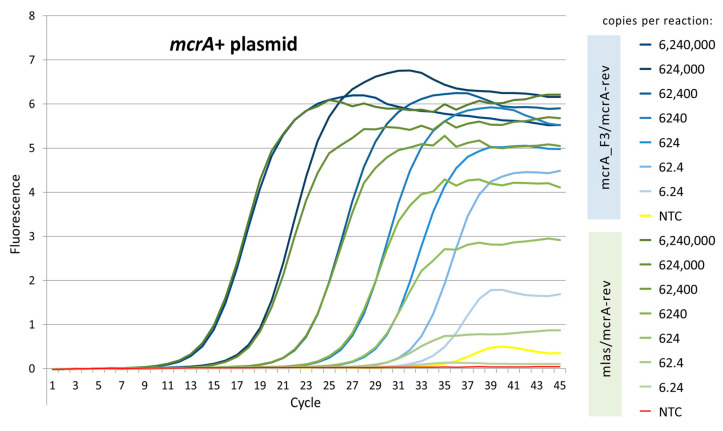
Comparison of amplification curves generated with annealing temperature of 55 °C.

**Figure 2 microorganisms-11-00660-f002:**
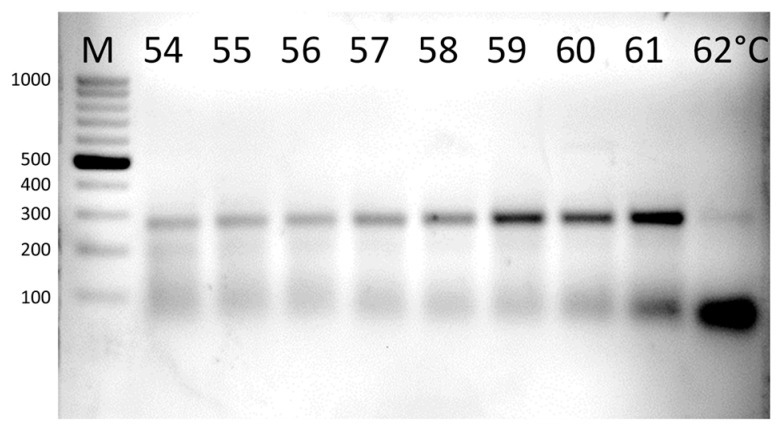
The results of gradient PCR. The optimal Ta was between 59 and 60 °C. The product length was 270 bp. M—DNA ladder.

**Figure 3 microorganisms-11-00660-f003:**
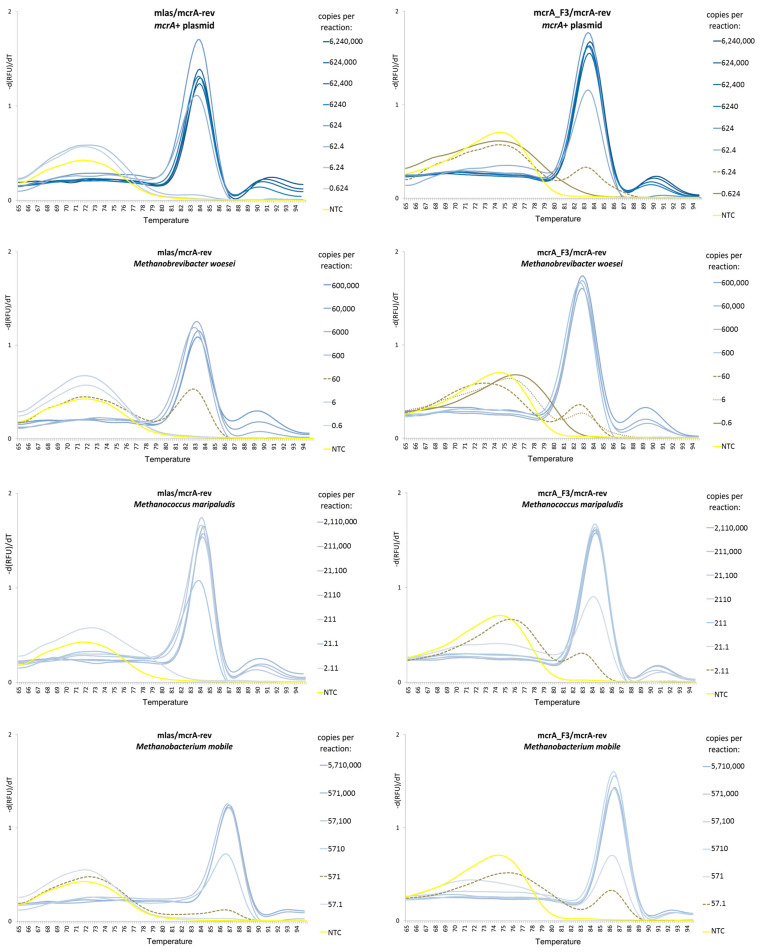
The melting profiles of 4 standards generated by using mlas/*mcrA*-rev and *mcrA*_F3/*mcrA*-rev primer pairs.

**Figure 4 microorganisms-11-00660-f004:**
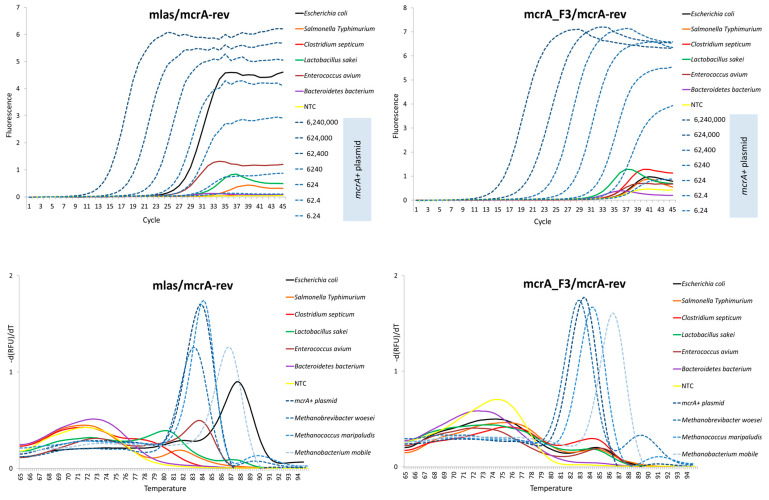
The melting profiles and the amplification plots of non-target controls set together with all 4 positive controls (upper charts) and with dilution series of the *mcrA*+ plasmid (lower charts) for comparison.

**Table 1 microorganisms-11-00660-t001:** The list of oligos used for the *mcrA* gene detection.

Oligo Name	Sequence 5′–3′	Oligo Binding Site 5′–3′ *	Tm [°C]	Product Size with Primer *mcrA*-Rev [bp]	References
*mcrA*-rev	CGTTCATBGCGTAGTTVGGRTAGT	446–467	62–68	n/a	[21]
mlas	GGTGGTGTMGGDTTCACMCARTA	1–23	62–70	469	[21]
*mcrA*_F3	CTTGAARMTCACTTCGGTGGWTC	199–221	62–66	271	This study and [14]

* Primer binding site refers to the nucleotide position in the *mcrA* gene of *Methanobrevibacter woesei* (acc. EU919432.1).

**Table 2 microorganisms-11-00660-t002:** Excerpt of the *mcrA* gene sequence alignment of methanogens in the binding sites of the newly designed *mcrA*_F3 primer.

Acc. No.	Species	*mcrA*_F3 Sequence CTTGAARMTCACTTCGGTGGWTC	Product Size with Primer *mcrA*-Rev (bp) *
KC618341.1	*Methanopyrus* sp.	ATGGAGACCCACTTCGGTGGATC	271
HQ896500.1	*Methanomassiliicoccus luminyensis*	ATGGAAACCCACTTCGGTGGTTC	271
AF414042.1	*Methanopyrus kandleri*	ATGGAGACGCACTTCGGAGGTTC	271
MH004454.1	*Methanosarcina mazei*	CTTGAAGACCACTTCGGTGGGTC	271
AB288266.1	*Methanosarcina horonobensis*	CTTGAAGACCACTTCGGTGGATC	271
AB288268.1	*Methanosarcina subterranea*	CTTGAAGACCACTTCGGTGGGTC	271
U22242.1	*Methanolobus oregonensis*	CTTGAGGACCACTTCGGTGGATC	n/a
EU715818.1	*Methanolobus zinderi*	CTCGAAGACCACTTCGGTGGATC	271
U22243.1	*Methanolobus taylorii*	CTCGAAGACCACTTTGGTGGATC	n/a
AB703629.1	*Methanolobus profundi*	CTCGAAGACCACTTCGGTGGATC	n/a
U22245.1	*Methanolobus vulcani*	CTCGAAGACCACTTCGGTGGATC	n/a
JN081865.1	*Methanocella conradii*	CTCGAGGACCACTTCGGCGGCTC	n/a
KJ441441.1	*Methanocella* sp.	CTCGAGGACCACTTCGGTGGCTC	n/a
AB300467.1	*Methanocella paludicola*	CTCGAGGACCACTTCGGTGGCTC	n/a
AF414044.1	*Methanomicrobium mobile*	ATGGAAGACCACTTCGGCGGTTC	n/a
MH004450.1	*Methanoculleus marisnigri*	ATGGAGGACCACTTCGGCGGTTC	271
AB300784.1	*Methanoculleus palmolei*	ATGGAGGACCACTTCGGCGGGTC	n/a
AB300787.1	*Methanoculleus bourgensis*	ATGGAAGACCACTTCGGCGGGTC	271
AB300779.1	*Methanoculleus chikugoensis*	ATGGAGGACCACTTCGGCGGTTC	271
DQ229160.1	*Methanogenium organophilum*	ATGGAAGACCACTTCGGTGGTTC	271
DQ229161.1	*Methanogenium boonei*	ATGGAAGACCACTTTGGCGGGTC	n/a
JQ917190.1	*Methanobacterium* sp.	CTGGAAGACCACTTTGGCGGTTC	n/a
GU385700.1	*Methanobrevibacter smithii*	CTTGAAACTCACTTCGGTGGATC	271
LK054628.1	*Methanobrevibacter oralis*	CTCGAAACTCACTTCGGTGGATC	n/a
CP001719.1	*Methanobrevibacter ruminantium*	CTTGAAACTCACTTCGGTGGTTC	271
KF214818.1	*Methanobrevibacter* sp. D5/*mcrA*+ plasmid	CTTGAAACTCACTTCGGTGGATC	271
EU919431.1	*Methanobrevibacter gottschalkii*	CTTGAAACTCACTTCGGTGGATC	271
KC865050.1	*Methanobrevibacter boviskoreani*	CTTGAAACCCAATTCGGTGGATC	271
EU919432.1	*Methanobrevibacter woesei*	CTTGAAACTCAATTCGGTGGATC	271
KC865051.1	*Methanobrevibacter wolinii*	CTTGAAACTCAATTCGGTGGATC	271
AF414035.1	*Methanobrevibacter arboriphilus*	CTTGAAACCCAATTCGGTGGTTC	n/a
HM802934.1	*Methanobacterium movens*	TTAGAAACCCTCTTCGGTGGATC	n/a
AY289750.1	*Methanobacterium thermaggregans*	CTCGAGGACCAGTTTGGTGGATC	n/a
AB300780.1	*Methanothermobacter wolfeii*	CTGGAGGACCAGTTCGGAGGATC	271
AB523786.1	*Methanothermobacter tenebrarum*	CTCGAGACACAATTTGGAGGATC	271
HQ283274.1	*Methanothermobacter crinale*	CTTGAGACACAGTTCGGCGGATC	n/a
KP006500.1	*Methanobacterium aggregans*	CTCGAGACCCAGTTCGGTGGATC	n/a
AY386125.1	*Methanobacterium aarhusense*	TTAGAAACACAGTTCGGTGGATC	n/a
AF313806.1	*Methanobacterium bryantii*	CTTGAAGATCAGTTCGGTGGATC	n/a
X07793.1	*Methanococcus voltae*	TTAGAAGACCACTTCGGTGGCTC	271
AF414048.1	*Methanothermococcus thermolithotrophicus*	TTAGAAGACCACTTCGGAGGTTC	271
AB703637.1	*Methanococcus maripaludis*	TTAGAAGACCACTTCGGTGGATC	271
M16893.1	*Methanococcus vannielii*	TTAGAAGACCACTTTGGTGGATC	271
AY354034.1	*Methanococcus aeolicus*	TTGGAAGACCACTTTGGAGGTTC	271
FJ982887.1	*Methanosphaera* sp.	TTAGAAGACCACTTCGGTGGATC	271
AF414047.1	*Methanosphaera stadtmanae*	TTAGAAGACCACTTTGGTGGATC	271

* n/a—not available due to incomplete *mcrA* sequence.

**Table 3 microorganisms-11-00660-t003:** Detection of the *mcrA* gene in both assays.

Estimated Plasmid/Genome Copy No.	mlas/*mcrA*-Rev			*mcrA*_F3/*mcrA*-Rev		
	No. Positive/No. Tested		Melting Analysis *	No. Positive/No. Tested		Melting Analysis *
***mcrA*^+^ Plasmid**						
**6.24 × 10^6^**	8/8		S	9/9		S
6.24 × 10^5^	8/8			9/9		
6.24 × 10^4^	8/8			9/9		
6.24 × 10^3^	8/8			9/9		
6.24 × 10^2^	8/8			10/10		
6.24 × 10^1^	8/8	6/8		10/10		
		2/8	S + PD			
6.24 × 10^0^	2/8		S + PD	3/10		S + PD
6.24 × 10^−1^	0/8		PD	0/10		PD
***Methanobrevibacter woesei* DSM 11979**						
6 × 10^6^	n/a		n/a	2/2		S
6 × 10^5^	8/8		S	7/7		
6 × 10^4^	8/8			7/7		
6 × 10^3^	8/8			7/7		
6 × 10^2^	8/8			8/8		
6 × 10^1^	8/8	2/8		8/8	4/8	
		6/8	S + PD		4/8	S + PD
6 × 10^0^	2/8		S + PD	2/8		S + PD
6 × 10^−1^	0/8		PD	0/8		PD
***Methanococcus maripaludis* DSM 2067**						
2.11 × 10^6^	8/8		S	9/9		S
2.11 × 10^5^	8/8			9/9		
2.11 × 10^4^	8/8			9/9		
2.11 × 10^3^	8/8			9/9		
2.11 × 10^2^	8/8			9/9		
2.11 × 10^1^	8/8	2/8		9/9	3/9	
		6/8	S + PD		6/9	S + PD
2.11 × 10^0^	2/8		S + PD	3/9		S + PD
2.11 × 10^−1^	0/8		PD	0/9		PD
***Methanomicrobium mobile* DSM 1539**						
5.71 × 10^6^	8/8		S	7/7		S
5.71 × 10^5^	8/8			7/7		
5.71 × 10^4^	9/9			7/7		
5.71 × 10^3^	8/8			7/7		
5.71 × 10^2^	9/9	1/9		7/7	4/7	
		8/9	S + PD		3/7	S + PD
5.71 × 10^1^	1/8		S + PD	7/7		S + PD
5.71 × 10^0^	0/8		PD	0/7		PD
5.71 × 10^−1^	0/8		PD	0/7		PD

* S = specific amplicons (mlas/*mcrA*-rev: 83.82 for *mcrA*+ plasmid, 83.27 for *Methanobrevibacter woesei*, 84.58 for *Methanococcus maripaludis*, and 87.08 °C for *Methanomicrobium mobile* amplicons; *mcrA*_F3/*mcrA*-rev: 83.55 for *mcrA*+ plasmid, 83.08 for *Methanobrevibacter woesei*, 84.65 for *Methanococcus maripaludis*, and 86.91 °C for *Methanomicrobium mobile* amplicons); PD = primer dimers.

**Table 4 microorganisms-11-00660-t004:** Reproducibility and efficiency of the assays.

Estimated Plasmid/Genome Copy No.	Mlas/*mcrA*-Rev					*mcrA*_F3/*mcrA*-Rev				
	Intra-Assay Variability		Inter-Assay Variability		Efficiency (%)	Intra-Assay Variability		Inter-Assay Variability		Efficiency (%)
	Mean Cq ± SD	CV%	Mean Cq ± SD	CV%		Mean Cq ± SD	CV%	Mean Cq ± SD	CV%	
***mcrA*+ Plasmid**										
6.24 × 10^6^	13.32 ± 0.02	0.11	13.20 ± 0.14	1.03	91.69	14.94 ± 0.17	1.10	15.11 ± 0.03	0.20	90.92
6.24 × 10^5^	17.09 ± 0.08	0.48	17.05 ± 0.35	2.05		19.16 ± 0.13	0.68	19.24 ± 0.19	0.97	
6.24 × 10^4^	21.08 ± 0.17	0.81	21.01 ± 0.47	2.24		23.48 ± 0.28	1.18	23.01 ± 0.4	1.76	
6.24 × 10^3^	24.72 ± 0.12	0.48	24.71 ± 0.24	0.96		27.32 ± 0.38	1.40	27.10 ± 0.29	1.05	
6.24 × 10^2^	28.26 ± 0.24	0.84	27.75 ± 0.55	1.97		30.91 ± 0.12	0.38	30.32 ± 0.8	2.64	
6.24 × 10^1^	30.58 ± 0.04	0.13	30.81 ± 0.57	1.85		33.77 ± 0.07	0.21	32.57 ± 0.77	2.36	
6.24 × 10^0^	31.56 * (1/3)	n/a	31.34 * (2/3) ± 0.31	0.99	n/a	34.55 * (1/3)	n/a	33.72 ± 1.05	3.11	n/a
6.24 × 10^−1^	negative					negative				
***Methanobrevibacter woesei* DSM 11979**										
6 × 10^5^	18.09 ± 0.21	1.14	18.45 ± 0.4	2.14	109.33	18.69 ± 0.34	1.81	19.62 ± 0.78	3.95	98.96
6 × 10^4^	21.80 ± 0.12	0.56	21.63 ± 0.41	1.89		22.95 ±0.2	0.88	23.33 ± 0.51	2.18	
6 × 10^3^	25.38 ± 0.3	1.17	25.29 ± 0.4	1.57		26.89 ± 0.18	0.68	26.72 ± 0.08	0.29	
6 × 10^2^	28.69 ± 0.13	0.45	28.43 ± 0.33	1.15		30.87 ± 0.11	0.36	30.53 ± 0.24	0.79	
6 × 10^1^	30.48 ± 0.15	0.50	30.63 ± 0.05	0.17		32.88 ± 0.3	0.92	32.76 ± 0.25	0.77	
6 × 10^0^	30.82 * (2/3) ± 0.21	0.69	30.97 * (1/3)	n/a	n/a	negative		33.90 * (2/3) ± 0.7	2.07	n/a
6 × 10^−1^	negative					negative				
***Methanococcus maripaludis* DSM 2067**										
2.11 × 10^6^	14.41 ± 0.25	1.72	14.79 ± 0.05	0.35	103.18	16.09 ± 0.52	3.22	16.80 ± 0.28	1.64	90.84
2.11 × 10^5^	18.07 ± 0.08	0.44	18.54 ± 0.04	0.24		20.30 ± 0.33	1.61	20.61 ± 0.15	0.71	
2.11 × 10^4^	22.53 ± 0.04	0.19	22.15 ± 0.3	1.37		24.26 ± 0.24	0.97	24.17 ± 0.31	1.29	
2.11 × 10^3^	25.98 ± 0.09	0.34	25.43 ± 0.52	2.06		28.14 ± 0.21	0.73	27.81 ± 0.63	2.27	
2.11 × 10^2^	29.21 ± 0.13	0.46	28.50 ± 0.61	2.14		31.81 ± 0.19	0.60	31.44 ± 0.44	1.40	
2.11 × 10^1^	30.84 ± 0.07	0.23	30.90 ± 0.17	0.55		34.87 ± 0.26	0.74	34.51 ± 0.81	2.36	
2.11 × 10^0^	negative		31.11 * (2/3) ± 0.06	0.18	n/a	negative		34.87 ± 0.59	1.68	n/a
2.11 × 10^−1^	negative					negative				
***Methanomicrobium mobile* DSM 1539**										
5.71 × 10^6^	18.47 ± 0.05	0.29	18.31 ± 0.21	1.12	104	20.19 ± 0.56	2.79	19.49 ± 0.78	4.01	97.67
5.71 × 10^5^	22.22 ± 0.03	0.11	22.00 ± 0.22	1.00		24.22 ± 0.3	1.24	23.31 ± 0.79	3.38	
5.71 × 10^4^	25.85 ± 0.09	0.33	25.44 ± 0.42	1.63		28.32 ± 0.14	0.48	27.38 ± 0.74	2.69	
5.71 × 10^3^	29.34 ± 0.2	0.69	28.68 ± 0.41	1.42		32.46 ± 0.26	0.79	30.54 ± 0.82	2.67	
5.71 × 10^2^	30.65 ± 0.3	0.97	31.12 ± 0.16	0.52		35.21 ± 0.59	1.69	33.86 ± 1.07	3.17	
5.71 × 10^1^	negative		31.73 * (1/3)	n/a	n/a	36.17 ± 0.79	2.18	36.19 ± 0.77	2.12	
5.71 × 10^0^	negative					negative				
5.71 × 10^−1^	negative					negative				

* instances where not all replicates tested positive; n/a—not applicable.

**Table 5 microorganisms-11-00660-t005:** The amplification results of the non-target controls.

DNA Template	Estimated Genome Copy No. Per Reaction or Amount *	Cq	
		Mlas/*mcrA*-Rev	*mcrA*_F3/*mcrA*-Rev
*Escherichia coli* ATCC 8739	4.38 × 10^5^ copies	26.76	34.17
*Salmonella* Typhimurium ATCC 14028	3.44 × 10^5^ copies	31.58	34.06
*Clostridium septicum* ATCC 12464	4.48 × 10^5^ copies	31.19	32.98
*Lactobacillus sakei* ATCC 15521	6.6 × 10^6^ copies	29.59	30.41
*Enterococcus avium* ATCC 14025	5.86 × 10^6^ copies	26.14	32.41
*Enterococcus casseliflavus* ATCC 700327	2.48 × 10^6^ copies	26.47	33.18
*Enterococcus raffinosus* ATCC 49464	7.7 × 10^4^ copies	27.65	33.20
*Blautia obeum* DSM 25238	2.08 × 10^7^ copies	28.11	34.28
*Ruminococcus gauvreauii* DSM 19829	8.83 × 10^5^ copies	27.82	35.21
*Helicobacter cinaedi* DSM 5359	1.71 × 10^7^ copies	24.63	36.36
*Desulfovibrio piger* DSM 749	1.32 × 10^8^ copies	29.06	34.49
*Proteus* sp.	6.1 ng	23.15	32.92
*Streptococcus* sp.	17 ng	26.45	34.66
*Streptococcus sp.* (beta-hemolytic)	0.118 ng	28.21	34.46
*Corynebacterium* sp.	1.15 ng	30.43	36.49
*Pseudomonas aeruginosa*	70 ng	26.35	32.14
*Pasteurella* sp.	0.377 ng	31.43	37.80
*Klebsiella* sp.	28 ng	26.23	32.45
*Staphylococcus sp.* (coagulase-negative)	3.28 ng	26.98	36.10
*Porphyromonas* sp.	2.43 ng	25.53	29.05
*Bacteroidetes* bacterium	1.13 ng	27.18	36.13

* fluorimetric measurements.

**Table 6 microorganisms-11-00660-t006:** Methanogens detected in chicken dropping samples with the use of mlas/*mcrA*-rev and *mcrA*_F3/*mcrA*-rev primer pairs.

Sample	Mlas/*mcrA*-Rev		*mcrA*_F3/*mcrA*-Rev	
	Cq	Tm (°C)	Cq	Tm (°C)
1	24.82	77.84; 82.25	28.17	82.51
2	24.54	71.84	29.62	79.47; 83.69
3	24.50	77.94; 83.23	23.17	83.36
4	25.59	83.18; 88.91	27.98	82.55
5	25.16	83.43	27.63	83.59
6	28.19	74.85	31.14	80.32; 83.28
7	28.09	77.62	30.63	83.05
8	negative	75.82; 80.96	28.27	77.08; 83.67
9	27.20	83.07	25.34	83.13
10	25.57	79.63; 83.17	28.15	83.11
11	25.73	80.65	28.58	77.11; 83.31
12	27.93	77.41	30.53	83.11
13	27.67	75.94; 82.02	28.80	82.88
14	28.36	82.97	29.40	77.51; 82.87
15	30.45	79.71	31.22	76.09; 82.87
16	25.60	75.51; 81.37	29.85	82.98
17	26.47	76.13; 81.10	28.90	76.59; 81.80
18	28.14	75.72	29.86	77.92; 83.12
19	29.29	75.91; 83.01	29.93	83.12
20	26.96	74.38	28.48	82.88

## Data Availability

Data are contained within this article and supplementary materials.

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
