# Peer review of "Improved Quantitative Real-Time PCR Protocol for Detection and Quantification of Methanogenic Archaea in Stool Samples"

_microorganisms, 2023, doi:10.3390/microorganisms11030660_

Round 1

Reviewer 1 Report

The manuscript by Cisek and colleagues describes a new set of primers to analyses methanogenic organisms from chicken droppings. To do so, 47 mrcA sequences were aligned and the primer designed and tested via qPCR. The paper is generally well written and reads well. It improves to detectability of methanogenic organisms. One point is in my opinion missing. The results from the chicken droppings were not sequenced, therefore one cannot really say that these primer pick up all (or more) methanogenic archaea than the previous primer set. An analysis like that would make this a well-rounded manuscript.

Otherwise I have only few minor comments.   

Comments:

Line 76: Table S1 should be with a small t to be concise

2.2 DNA Templates: Did you cultivate the organisms or did you receive the purified DNA from somewhere? If you cultivated them, please briefly indicate growth conditions, time etc.

Line 89: “binding site”

Table 6: Did you ever sequence the PCR products from the chicken droppings to determine the actual diversity, which is recovered with your primers? This would make a valuable contribution to the manuscript.

Author Response

Dear Reviewer,

thank you for your thorough review and valid remarks. Changes have been made accordingly.

Remark 1: Line 76: Table S1 should be with a small t to be concise

Response: It has been fixed.

Remark 2: 2.2 DNA Templates: Did you cultivate the organisms or did you receive the purified DNA from somewhere? If you cultivated them, please briefly indicate growth conditions, time etc.

Response: We did not culture any of the microorganisms. They were bought/received either in a form of DNA, or bacterial pellet from which we isolated DNA with the use of Genomic Mini AX Bacteria+ kit (A&A Biotechnology, Gdynia, Poland). This information is now further specified in the manuscript.

Remark 3: Line 89: “binding site”

Response: It has been fixed.

Remark 4: Table 6: Did you ever sequence the PCR products from the chicken droppings to determine the actual diversity, which is recovered with your primers? This would make a valuable contribution to the manuscript.

Response: We have attempted to sequence the PCR products from the chicken droppings. Briefly, after real-time PCR, the products were separated on gel electrophoresis and the bands representing the specific products were cut out and purified. Then the products were subjected to the Sanger sequencing. However, since the chromatograms showed a lot of background noise, the one thing we could say for sure was that there was more than one species of methanogens in these samples. Nonetheless, we have performed the BLAST analysis, which showed that these products were most closely related to Methanobrevibacter ruminatum. But we still don’t know what other background species were present in these samples, therefore we chose not to include this information in the manuscript.

In order to avoid this problem, we would have to perform a more detailed sequencing of the entire population of amplificons generated with our mcrA-targeting primers. Unfortunately, we did not have enough funds to do so.

Reviewer 2 Report

My comments are within the manuscript. 

The results are well wrote and I would thank the author for this work because as I work on methanogen in human microbiota, we really need a specific primers to screen a sample before culture. 

regarding the good work presented here, the PCRs systems designed will be usefull for scientifics work on methanogens.

Author Response

Dear Reviewer,

thank you for your thorough review of our manuscript. We have made the following changes.

Remark 1: Line 25 - reference.

Response: The following references have been added:

  • Djemai, K.; Drancourt, M.; Tidjani Alou, M. Bacteria and Methanogens in the Human Microbiome: A Review of Syntrophic 347 Interactions. Microb. Ecol. 2022, 83, 536–554, doi:10.1007/S00248-021-01796-7.
  • Li, Z., Wang, X., Zhang, T., Si, H., Xu, C., Wright, A.G., Li, G. Heterogeneous development of methanogens and the correlation with bacteria in the rumen and cecum of sika deer (Cervus nippon) during early life suggest different ecology relevance. BMC Microbiol. 2019, 19(1), 129, doi: 10.1186/s12866-019-1504-9.

Remark 2: I found these publications on the presence of methanogens in chicken ceca:

https://ami-journals.onlinelibrary.wiley.com/doi/full/10.1111/j.1472-765X.2007.02243.x

https://www.researchgate.net/publication/6710832_Identification_and_Quantification_of_Methanogenic_Archaea_in_Adult_Chicken_Ceca

Could you please why did you used chicken ceca sample.

Response: Perhaps we were not precise, since yes, there are some publications reporting methanogens in chicken ceca. We have now changed this sentence into “For instance, to date many studies do not report the presence of archaea in the chicken ceca at all [Medvecky et al. 2018; Segura-Wang et al. 2021], whilst other say otherwise [Saengkerdsub et al. 2007a; Saengkerdsub 2007b]” and added more references:

  • Medvecky, M.; Cejkova, D.; Polansky, O.; Karasova, D.; Kubasova, T.; Cizek, A.; Rychlik, I. Whole Genome Sequencing and Function Prediction of 133 Gut Anaerobes Isolated from Chicken Caecum in Pure Cultures. BMC Genomics 2018, 19, doi:10.1186/S12864-018-4959-4
  • Segura-Wang, M.; Grabner, N.; Koestelbauer, A.; Klose, V.; Ghanbari, M. Genome-Resolved Metagenomics of the Chicken Gut Microbiome. Front. Microbiol. 2021, 12, 2390, doi:10.3389/FMICB.2021.726923/BIBTEX.
  • Saengkerdsub, S., Herrera, P., Woodward, C.L., Anderson, R.C., Nisbet, D.J., Ricke, S.C. Detection of methane and quantification of methanogenic archaea in faeces from young broiler chickens using real-time PCR. Lett Appl Microbiol. 2007, 45(6), 629-34, doi: 10.1111/j.1472-765X.2007.02243.x.
  • Saengkerdsub, S., Anderson, R.C., Wilkinson, H.H., Kim, W.K., Nisbet, D.J., Ricke, S.C. Identification and quantification of methanogenic Archaea in adult chicken ceca. Appl Environ Microbiol. 2007, 73(1), 353-6, doi: 10.1128/AEM.01931-06.

We used chicken droppings because we knew that chickens in that precise flock were healthy, housed in a well-maintained henhouse, and had access to various food sources and an outdoor environment, from which they could potentially acquire the methanogenic archaea. Moreover, it was a mature flock, with new chicks being bought and introduced into the coop from time to time, in place of the old ones. Therefore, in general, we were almost certain that these chickens should be colonized by methanogens of all other sources available to us. In addition, for technical and administrative reasons, the droppings were more accessible to us than ceca.

Remark 3: Add amplicon size.

Response: A column with amplicon sizes has been added to table 2.

Remark 4: Write the gene name in italic within the text.

Response: All gene names are now in italic.